# New Species and New Records of *Otidea* from China Based on Molecular and Morphological Data

**DOI:** 10.3390/jof8030272

**Published:** 2022-03-08

**Authors:** Yu-Yan Xu, Ning Mao, Jia-Jia Yang, Li Fan

**Affiliations:** College of Life Science, Capital Normal University, Xisanhuanbeilu 105, Haidian, Beijing 100048, China; 2190801004@cnu.edu.cn (Y.-Y.X.); 2210801013@cnu.edu.cn (N.M.); 2210802122@cnu.edu.cn (J.-J.Y.)

**Keywords:** Ascomycota, Pyronemataceae, phylogeny, seven new taxa, taxonomy

## Abstract

Species of genus *Otidea* previously reported in China are mainly distributed in the northeast, northwest and southwest regions of China, but the species diversity of *Otidea* in north China is not very clear. In this study, newly collected *Otidea* specimens from northern China and some herbarium specimens deposited in three important Chinese fungus herbaria (HMAS, HKAS, HMJAU) were studied using morphological and phylogenetic methods. The internal transcribed spacers of the nrDNA (ITS), the nrRNA 28S subunit (nrLSU), the translation elongation factor 1-alpha (*tef1-**α*), and the second largest subunit of RNA polymerase II (*rpb2*), were employed to elucidate the phylogenetic relationships between *Otidea* species. Results identified 16 species of *Otidea*, of which seven new species are described, namely *O.*
*aspera*, *O.*
*cupulata*, *O. filiformis*, *O.*
*khakicolorata*, *O. parvula*, *O.*
*plicara* and *O.*
*purpureobrunnea*. *Otidea bicolor* and *O. pruinosa* are synonymized as *O. subpurpurea*. Two species, *O. mirabilis* and *O. nannfeldtii*, are being reported for the first time in China. The occurrence of *O. bufonia*, *O. leporina* and *O. onotica* are confirmed by molecular data in China.

## 1. Introduction

The genus *Otidea* (Pers.) Bonord. (Pyronemataceae, Pezizales), with *O. onotica* (Pers.) Fuckel as the type species, was established in the mid-19th century [1,2]. The genus is characterized by the following criteria: epigeous, cup- to ear-like apothecia, split to the base on one side (less often entirely), and being stipitate or not; or as in a single species, hypogeous with enclosed ascomata, subcylindrical, nonamyloid, operculate asci, ellipsoid to fusoid, biguttulate ascospores, and filiform paraphyses [3,4,5]. *Otidea* species are considered to form ectomycorrhizae with both broad leaf and conifer trees [3,4], and are widely distributed across the temperate regions of Europe, North America and Asia in the northern hemisphere [4,5,6,7,8,9,10]. The taxonomy of *Otidea* species are mainly based on the morphological characteristics studied in work before 2006 [5,6,11,12,13,14,15,16,17]. Since then, taxonomists began to introduce molecular methods into the taxonomy and identification of *Otidea* species. Molecular techniques have revolutionized phylogenetics and species delimitation of *Otidea* [3,4,10,18,19,20,21,22,23,24,25]. In 2009, a whole new set of morphological and histochemical features was also introduced in *Otidea* by Harmaja [15] and further by Hansen and Olariaga [3] and Olariaga et al. [4]. These features have made it possible to recognize the species morphologically to a large extent and have since been employed. *Otidea* species in Europe (c. 32 accepted species) and North America (c. 14 accepted species) have been comprehensively and systematically studied in recent years [3,4,10,21,22].

China has a huge temperate area in the northern hemisphere and likely has diverse *Otidea* species. However, only a few taxonomic works focus on this genus, and about a quarter of Chinese *Otidea* species are not supported by molecular evidence [6,8,16,17,18,23,24,25,26,27,28,29]. Currently, a total of 24 *Otidea* species are reported in China, including 17 native species and seven known species originally described from Europe and/or North America.

During our investigation of fungal resources in northern China since 2017, many apothecia of the genus *Otidea* were collected. On the bases of these new collections and some of herbarium specimens deposited in three important Chinese fungus herbaria (HMAS, HKAS, HMJAU), we recognized seven species and two records new to China based on both morphological examination and molecular analysis. Also, both *O**. bicolor* W.Y. Zhuang & Zhu L. Yang, and *O. pruinosa* Ekanayaka, Q. Zhao & K.D. Hyde were conspecific with *O. subpurpurea* W.Y. Zhuang. Our aim in this paper is to describe and illustrate these new species and new records and to synonymize *O**. bicolor* and *O. pruinosa* as *O. subpurpurea*. Molecular data for some known species existing in China are additionally provided.

## 2. Materials and Methods

### 2.1. Morphological Studies

Fresh specimens were collected and photographed in the field from the Shanxi and Hebi provinces, as well as Beijing, China. The specimens were dried and deposited in the BJTC (Herbarium, Biology Department, Capital Normal University) and the HSA (Herbarium Institute of Edible Fungi, Shanxi Academy of Agricultural Science, Taiyuan, China). Other specimens were studied from the HMAS (Herbarium Mycologicum Academiae Sinicae, Institute of Microbiology, Chinese Academy of Sciences), HKAS (Herbarium of Cryptogams at the Kunming Institute of Botany, Chinese Academy of Sciences) and HMJAU (Herbarium of Mycology, Jilin Agricultural University). Macroscopic characteristics were recorded from fresh specimens. Standardised color values matching the described colour were taken from ColorHexa (http://www.colorhexa.com/, accessed on 30 January 2022). Microscopic characteristics were observed in thin sections of dry specimens mounted in 5% KOH and Melzer’s reagent [30]. The dimensions for ascospores are given using notation of the form (a-)b-c(-d). The range b-c contains a minimum of 90% of the measured values. The extreme values, i.e., a and d, are given in the parentheses. L_m_ and W_m_ indicate the average ascospore length and width for the measured ascospores, respectively. Q is used to represent length/width ratio of a ascospore in side view and Q_m_ represents average Q of all specimens. The number of populations that the statistics are based on is indicated by n.

### 2.2. DNA Extraction, PCR Amplification, Sequencing

Herbarium specimens were crushed by shaking for 30 s at 30 Hz 2–4 times (Mixer Mill MM 301, Retsch, Haan, Germany) in a 1.5 mL tube together with one 3 mm diameter tungsten carbide ball, and total genomic DNA was extracted using the modified CTAB method [31]. The following primers were used for PCR amplification and sequencing: ITS1f/ITS4 [31,32] were used for the internal transcribed spacers of the nrDNA (ITS1-5.8S-ITS2 = ITS), LR0R/LR5 [33] for the nrDNA 28S subunit (nrLSU), EF1-983F/EF1-2218R [34] for the translation elongation factor 1-alpha (*tef1-α*), and RPB2-Otidea6F/RPB2-Otidea7R and fRPB2-7cF/fRPB2-11aR [3] for the RNA polymerase II second largest subunit (*rpb2*), respectively. PCRs were performed in 50 μL reactions containing 4 μL DNA template, 2 μL of per primer (10 μM), 25 μL 2× Master Mix (Tiangen Biotech Co., Beijing, China), and 17 μL ddH_2_O. PCR reactions were performed as follows: for the ITS gene: initial denaturation at 94 °C for 3 min, followed by 35 cycles at 94 °C for 30 s, 56 °C for 45 s, 72 °C for 1 min, and a final extension at 72 °C for 10 min; for the nrLSU gene: initial denaturation at 94 °C for 4 min, followed by 35 cycles at 94 °C for 30 s, 55 °C for 45 s, 72 °C for 1 min, and a final extension at 72 °C for 10 min; for the *tef1-α* gene: initial denaturation at 94 °C for 3 min, followed by 35 cycles at 94 °C for 30 s, 60 °C for 45s, 72 °C for 1min, and a final extension at 72 °C for 10 min; for the *rpb2* gene: initial denaturation at 94 °C for 3 min, followed by 10 cycles (including denaturation) at 94 °C for 30 s, annealing temperature started at 62 °C (decreased by 1 °C per cycle, until to 52 °C) for 45 s and extension at 72 °C for 1 min, then followed by 30 cycles at 94 °C for 35 s, 55 °C for 45 s, 72 °C for 1 min, and a final extension at 72 °C for 10 min. The PCR products were sent to Beijing Zhongkexilin Biotechnology Co., Ltd. (Beijing, China) for purification, sequencing, and editing. The newly generated sequences were assembled and edited using SeqMan (DNA STAR package, DNAStar Inc., Madison, WI, USA) with generic-level identities for sequences confirmed via BLAST queries of GenBank.

### 2.3. Sequence Alignment and Phylogenetic Analyses

A total of 730 sequences from 283 collections of *Otidea* were used in the molecular phylogenetic analyses. The detail information about them is provided in Appendix A, including the geographic origin and accession numbers. Sequences of all DNA regions generated in this study were deposited in GenBank. The sequences obtained from GenBank are based on published literature or selected by using BLASTn search to find similar matches with taxa in *Otidea*. Two datasets were assembled for this study. Dataset I (ITS/nrLSU) and datasets II (ITS/nrLSU/*tef1-α*/*rpb2*) contained the backbone species and all phyloclades of *Otidea*, which were used to infer the phylogenetic status of Chinese *Otidea* species of the genus *Otidea*. The taxa *Monascella botryosa* Guarro & Arx and *Warcupia terrestris* Paden & J.V. Cameron were selected as outgroups. The ITS, nrLSU, *tef1-α* and *rpb2* sequences were respectively aligned using the MAFFT v.7.110 online program under the default parameters [35], and manually adjusted to allow for maximum sequence similarity in Se-Al version.2.03a [36]. Ambiguously aligned regions of each sequence were detected and excluded using Gblocks 0.91b [37] before the phylogenetic analyses. Unsampled gene regions were coded as missing data and all introns of *tef1*-*α* and *rpb2* were excluded because of the alignment difficulty. To examine the conflict among topologies with maximum likelihood (ML), separate single-gene analyses were conducted. Alignments were concatenated using SequenceMatrix v1.7.8 [38] and are provided in Appendix A. We conducted maximum likelihood (ML) and Bayesian inference (BI) analyses on the two datasets.

Maximum likelihood (ML) analyses of the two datasets were carried out using RAxML 8.0.14 [39] with all parameters kept to the default settings using a GTRGAMMAI model. The ML bootstrap replicates (1000) were computed in RAxML using a rapid bootstrap analysis searching for the best scoring ML tree. Bayesian inference (BI) analyses were performed with MrBayes v3.1.2 [40] based on the best substitution models for each gene region as determined by MrModeltest 2.3 [41]. The GTR + I+G model was the best model for ITS, nrLSU and *rpb2*, whereas the best model for *tef1-α* was the SYM + I+G model. Two independent executions of four chains were conducted: 3,485,000 for ITS/nrLSU and 765,000 for the ITS/nrLSU/*tef1-α*/*rpb2* datasets. Markov chain Monte Carlo generations were conducted using the default settings and sampled every 100 generations. The temperature value was lowered to 0.20, burn-in was set to 0.25, and the program was automatically stopped as soon as the average standard deviation of split frequencies reached below 0.01. A 50% majority-rule consensus tree was constructed. Clades with a bootstrap support (BS) ≥ 70% and a Bayesian posterior probability (PP) ≥ 0.95 were considered as significantly supported [42,43]. All phylogenetic trees were viewed with TreeView32 [44].

## 3. Results

### 3.1. Phylogenetic Analyses

No topological incongruence was detected when the four genes were analyzed individually. Dataset I (ITS/nrLSU) contained 528 sequences from 51 species, including 93 novel sequences the two genes from Chinese collections, and four from the outgroups (*Monascella botryosa* and *Warcupia terrestris*). The dataset had an aligned length of 1363 characters (551 bp from ITS and 812 bp from nrLSU), of which 647 were constant, 716 were variable, and 609 of these variable sites were informative. ML and BI analyses yielded similar tree topologies. Only the tree inferred from the ML analyses is shown (Figure 1). The species of *Otidea* formed a monophyletic clade with high support values (BS = 100%, PP = 1.00). A total of 10 clades were recognized in the two-gene phylogram, which is consistent with Olariaga et al. [4] and Hansen and Olariaga [3]. Newly obtained Chinese *Otidea* specimens were nested in six clades: *O. bufonia-onotica*, *O. formicarum*, *O. leporina*, *O. cantharella*, *O. alutacea*, and *O. platyspora* clade (Figure 1). A total of 16 species were recognized.

In the *O. bufonia-onotica* clade, 24 Chinese specimens were clearly placed in eight well-supported clades, represented by five known species and three new species. The known species are *O. bufonia* (Pers.) Boud., *O. subpur**purea*, *O. mirabilis* Bolognini & Jamoni, *O. onotica* and *O. brevispora* (W.Y. Zhuang) Olariaga & K. Hansen. The three new species are respectively described as *O. filiformis*, *O.*
*cupulata* and *O.*
*purpureobrunnea* in this study. It is interesting that the sequences from the type specimens of *O. bicolor* and *O. pruinosa* fall into the clade of *O. subpur**purea* and shared 98.87–99.84% similarity in the ITS region, which implied they may be conspecific, although they have some difference in apothecial color. Fortunately, we borrowed the type specimens of these three species for observation and research. We thought that they should be conspecific and formally synonymized *O. bicolor* and *O. pruinosa* in this study.

In the *O. formicarum* clade, seven Chinese specimens were clearly placed in two well supported clades. One of clades corresponds to *O. nannfeldtii* Harmaja, marking the first time it is discovered in China. The other one is a new species described as *O.*
*khakicolorata* in this study. In the *O. leporina* clade and *O. cantharella* clade, the Chinese specimens were identified as the previously known species, *O. leporina* (Batsch) Fuckel and *O. propinquata* (P. Karst.) Harmaja, respectively. In the *O. alutacea* clade, seven Chinese collections were clearly placed into three well supported clades, represented by the known species *O. alutacea* (Pers.) Massee and two new species described as *O. parvula* and *O.*
*aspera* in this paper. In the *O. platyspora* clade, the two Chinese collections formed a distinct clade with high evidential support, which was described as *O.*
*plicara* in this study.

In order to further verify the phylogenetic positions of the seven new species and whether *O. bicolor* and *O. pruinosa* are conspecific with *O. subpur**purea*, we performed a further phylogenetic analysis based on the four-gene dataset II. This dataset II does not include sequences from seven confirmed known species, namely *O*. *bufonia*, *O*. *mirabilis*, *O*. *onotica*, *O*. *nannfeldtii*, *O*. *leporina*, *O*. *propinquata* and *O*. *alutacea*. Dataset II (ITS/nrLSU/*tef1-α*/*rpb2*) contained 501 sequences from 49 species, including 73 novel sequences these four genes from Chinese collections, and 8 from the outgroups (*M**. botryosa* and *W**. terrestris*). The dataset had an aligned length of 4400 characters (551 bp from ITS, 812 bp from nrLSU, 1383bp from *tef1-α* and 1654 bp from *rpb2)*, of which 2456 were constant, 1854 were variable, and 1753 of these variable sites were informative. ML and BI analyses yielded similar tree topologies. Only the tree inferred from the ML analysis is shown (Figure 2).

The species of *Otidea* formed a monophyletic clade with high support values (BS = 100%, PP = 1.00). A total of 10 clades were recognized in the four-gene phylogram, which is consistent with Hansen and Olariaga [3]. Similar to the two-gene phylogram results, the specimens from China formed seven apparently independent clades with high support values, representing seven new species. The type sequences of *O. bicolor*, *O. pruinosa* and *O. subpurpurea* clustered into a clade with high support values (BS = 100%, PP = 1.00), indicating that they are indeed the same species, and *O. bicolor* and *O. pruinosa* are placed in synonymy as *O. subpurpurea* in the taxonomy section below. Compared with the two-gene tree phylogram (Figure 1), the 10 clades identified in the four-gene tree are consistent, but inside the *O. bufonia-onotica* clade, the *O. concinna* clade, and the *O. alutacea* clade, the topological positions of some species are slightly different (e.g., *O. purpureobrunnea* and *O. cupulata* in the *O. bufonia-onotica* clade), which may be due to the fact that these species have a lower support value between them.

### 3.2. Taxonomy

Based on our phylogenies and morphological data, a total of seven new species, a known species, and two new records of *Otidea* from China were described and illustrated here.

***Otidea******aspera*** L. Fan & Y.Y. Xu, sp. nov. (Figure 3).

MycoBank: MB843176.

Etymology: *aspera*, referring to the rough receptacle surface.

Holotype: China. Shanxi Province, Jiaocheng County, Pangquangou National Nature Reserve, HaoJiagou, alt. 1800m, in the mixed forest dominated by *Pinus tabuliformis* Carrière and *Quercus mongolica* Fisch. ex Ledeb., 29 August 2018, J.Z. Cao, LH278 (HSA 278).

Saprobic on soil. Apothecia solitary or caespitose in nature, 30–45 mm high, 25–60 mm wide, initially ear-shaped, then expanding and sometimes becoming irregularly ear-shaped or shallowly to deeply cup-shaped, sometimes elongated on one side or obconical, split, stipitate. Hymenium surface greyish yellow (#fbe8c5) to light brown (#baab98) when fresh, ochre brown (#a87832) when dry, subsmooth. Receptacle surface pale yellow (#fafad2) to yellowish brown (#b5a27f) when fresh, slightly hygrophanous, pale whitish ochre (#c8b99f) when dry, finely warty. Stipe 5–12 × 4–8 mm. Basal tomentum and mycelium white. Apothecial section 600–900 µm thick. Ectal excipulum of *textura angularis*, 100–150 µm thick, cells thin walled, hyaline to pale brown, 11–28 × 8–20 µm. Medullary excipulum of *textura intricata*, 300–500 µm thick, hyphae 3–10 µm wide, sometimes slightly swollen, thin to thick walled, septate, hyaline to light brown. Subhymenium c. 50–120 µm thick, visible as a brown zone of densely arranged cylindrical to swollen cells. Paraphyses septate, straight to slightly curved, of uniform width or slightly enlarged at the apices to 3–4.7 µm wide, without or rarely with 1–2 low notches. Asci 150–200 × 9–13 µm, 8-spored, unitunicate, cylindrical, hyaline, long pedicellate, arising from croziers, non-amyloid, ascospores released from an eccentric split at the apical apex. Ascospores ellipsoid to slightly subfusoid, inequilateral, with two large guttules, sometimes only with one big guttule, smooth, hyaline, (12–) 12.8–15 (–15.5) × (5.8–) 6.5–7.5 (–8) µm (L_m_ × W_m_ = 14 × 7 µm, Q = 1.8–2.2, Q_m_ = 2, n = 50). Receptacle surface with warts, 50–80 µm high, formed by short, fasciculate, hyphoid hairs, of 5–7 subglobose to elongated cells, constricted at septa, 6–13 µm wide. Resinous exudates absent to scarce. Basal mycelium of 2–5 µm wide, septate, hyaline to pale brown hyphae, smooth, unchanged in KOH, smooth, turning yellow in MLZ.

Other materials examined: China. Inner Mongolia Autonomous Region, Daqinggou National Nature Reserve, on the broad-leaved woodland, 24 August 2005, Tolgor Bau (HMJAU 4166).

Notes: *Otidea aspera* is diagnosed by the combination of the stipitate, broadly ear shaped to cup-shaped, greyish yellow to light brown hymenium, pale-yellow to yellowish-brown receptacle surface, ellipsoid to slightly subfusoid ascospores and straight to slightly curved paraphyses. *Otidea aspera* and *O. parvispora* have comparable apothecia color, however *O. parvispora* differs from *O. aspera* by the smaller ascospores ((11.0–) 11.5–13.0 × 5.0–6.5 µm) and shorter asci. DNA analysis showed that *O. aspera* shared less than 93.39% ITS sequence similarity with other *Otidea* species. Phylogenetic analyses revealed that the sequences of *O. aspera* were grouped into an independent clade with a strong support value (Figure 1 and Figure 2). These supported the erection of the new species.

***Otidea cupulata*** L. Fan & Y.Y. Xu, sp. nov. (Figure 4).

MycoBank: MB843177.

Etymology: *cupulata*, referring to the apothecia shape of the fungus.

Holotype: China. Shanxi Province, Jiaocheng County, Pangquangou Township, Badaogou valley, alt. 2200m, on soil in mixed forest of *Larix* sp. and *Betula* sp., 28 August 2018, L.J. Guo, LH218 (HSA 218).

Saprobic on soil. Apothecia gregarious or caespitose in nature, 25–40 mm high, 15–50 mm wide, initially ear-shaped, soon expanding, becoming broadly ear-shaped or deeply cup-shaped, split, often broader above, margin sometimes lobate, stipitate. Hymenium surface dark orange brown (#734a12) to dark purple brown (#4d282d) when fresh, and usually with bluish-lilaceous shades, gray ochraceous brown (#37290e) when dry, subsmooth. Receptacle surface yellowish brown (#a1805f) to brown (#8a6660) when fresh, slightly hygrophanous, surface with shallow wrinkles, dark brown (#261600) when dry, furfuraceous. Stipe 5–10 × 5–7 mm. Basal tomentum and mycelium whitish to grayish yellow (#c6cbac). Apothecial section 900–1200 µm thick. Ectal excipulum of *textura angularis*, 80–110 µm thick, cells thin walled, brownish, 10–25 × 8–22 µm. Medullary excipulum of *textura intricata*, 500–700 µm thick, hyphae 3.5–10 µm wide, sometimes slightly swollen, thin to slightly thick walled, septate, hyaline to light brown. Subhymenium c. 60–100 µm thick, visible as a brown zone, of densely arranged cylindrical to swollen cells, with scattered brown resinous exudate at septa. Paraphyses septate, curved to hooked of uniform width or slightly enlarged at the apices to 2.6–4.2 µm wide, without or rarely with 1–2 low notches, sometimes forked near the apex. Asci 150–200 × 8–11 µm, 8-spored, unitunicate, cylindrical, hyaline, long pedicellate, arising from croziers, non-amyloid, ascospores release from an eccentric split at the apical apex. Ascospores ellipsoid to slightly subfusoid, inequilateral, with two large guttules, sometimes with only one big guttule, smooth, hyaline, (12–) 12.5–15 (–15.5) × (6–) 6.5–7.4 (–7.9) µm (L_m_ × W_m_= 13.9 × 7 µm, Q = 1.9–2.1, Q_m_ = 2, n = 50). Receptacle surface with low warts, 25–45 µm high, formed by short, fasciculate, hyphoid hairs, of 3–4 subglobose to elongated cells, constricted at septa, 6–11 µm wide, sometimes with a gelatinous sheath. Resinous exudates abundant on the outer surface, dark brown, partly dissolving and converting into small particles in MLZ, entirely dissolving and turning bright yellow in KOH. Basal mycelium of 3–6.5 µm wide, septate, hyaline to pale brown hyphae, bright yellow in KOH, with abundant, very small, irregularly, brown, resinous exudates on the surface, dissolving and turning bright yellow in KOH, unchanged in MLZ.

Other materials examined: China. Shanxi Province, Jiaocheng County, Pangquangou Township, Badaogou Valley, alt. 1800m, on soil under *Larix* sp., 6 September 2018, L.J. Guo, LH 406 (HSA 406).

Notes: *Otidea cupulata* is recognized by the stipitate, broadly cup-shaped, dark-orange-brown to dark-purple-brown hymenium, yellowish-brown to brown receptacle surface, ellipsoid to subfusoid ascospores, forked or notched paraphyses, and furfuraceous receptacle surface. The forked paraphyses is rarely found in other species in the O. bufonia-onotica clade. Several species in the O. bufonia-onotica clade are similar to *O. cupulata* in apothecial shape and color, including *O. bufonia*, *O. filiformis*, *O. mirabilis,* and *O. olivaceobrunnea* Harmaja, but *O. bufonia* can be distinguished by its distinctly narrowly fusoid ascospores, the presence of hyphae with striate resinous exudates in the medullary excipulum, and resinous exudates of the ectal excipulum that does not turn bright yellow in KOH. *Otidea filiformis* differs in its apothecia with pinkish shades, distinctly narrowly fusoid ascospores, unenlarged and curved to hooked paraphyses, and higher warts (40–75 µm) on the receptacle surface. *Otidea mirabilis* differs by having purple to lilaceous-bluish shades on the receptacle surface and narrowly fusoid ascospores (Q_m_ = 2.1–2.3). *Otidea olivaceobrunnea* can be separated by its olive-brown hymenium and wider ascospores (14–17 × 8–8.5 µm). Four species (*O. purpureogrisea* Pfister, F. Xu & Z.W. Ge, *O. purpureobrunnea*, *O. simithii* Kanouse and *O. subpurpurea*) have more or less bluish or lilaceous shades on apothecia that resembles that of *O. cupulata*. However, *O. purpureogrisea* is distinguished by its ear-shaped apothecia, dark-purple-brown to purple-gray receptacle surface, and the resinous exudate in the ectal excipulum turning amber and brown in KOH. *Otidea purpureobrunnea* is distinguished by its grayish-purple-brown to dark-purple-brown receptacle surface and mostly smooth basal mycelium. *Otidea simithii* differs in having typically narrower, ear-shaped apothecia, and resinous exudates of the ectal excipulum that does not turn bright yellow in KOH. *Otidea subpurpurea* in having smaller ascospores (9–12 × 4.5–6 µm) and lilac to purplish receptacle surface.

In the two-gene phylogenetic tree (Figure 1), two specimens (K(M)41595 and K(M)156077) from England fall into the clade of *O*. *cupulata*, which are noted to belong to an unnamed taxon related to *O. bufonia* and *O. subpurpurea* by Parslow et al. [10]. Regrettably, they did not further describe the morphology of these two specimens. According to the ITS sequence similarity analysis (ITS: 98%) and phylogenetic analyses (Figure 1), they may be conspecific with *O.*
*cupulata*. Final confirmation requires morphological observation of these two specimens.

***Otidea filiformis*** C.L. Hou, Y.Y. Xu & H. Zhou, sp. nov. (Figure 5).

MycoBank: MB843178.

Etymology: *filiformis*, referring to the filiform paraphyses in the hymenium.

Holotype: China. Beijing City, Huairou District, Sunshanzi Village, alt. 770m, on soil in mixed forest of *Populus* sp. and *Larix* sp., 28 August 2020, G.Q. Chen C505 (BJTC C505).

Saprobic on soil. Apothecia gregarious or caespitose in nature, 15–55 mm high, 15–55 mm wide, initially ear-shaped, soon expanding, becoming shallowly or deeply cup-shaped, split, margin sometimes lobate, sessile or shortly stipitate, regular or sometimes undulate in the margin. Hymenium surface yellowish brown (#b19461) to ochre yellow (#cd9575) with pinkish shades, sometimes with brown spots or stains when fresh, margin dark brown when bruised, gray brown (#321f15) when dry, subsmooth. Receptacle surface yellowish brown (#b19461) to orange brown (#967059) when fresh, slightly hygrophanous, dark brown (#321f15) when dry, furfuraceous to finely warty. Stipe not well developed. Basal tomentum and mycelium whitish to pale brown (#dccdbf). Apothecial section 800–1000 µm thick. Ectal excipulum of *textura angularis*, 75–110 µm thick, cells thin walled, brown, 10–35 × 7–26 µm. Medullary excipulum of *textura intricata*, 400–500 µm thick, hyphae 4–10 µm wide, sometimes slightly swollen, thin to thick walled, septate, hyaline to light brown, without resinous exudates. Subhymenium ca. 75–100 µm thick, visible as a brown zone of densely arranged cylindrical to swollen cells, with scattered brown resinous exudate at septa. Paraphyses septate, curved to hooked of uniform width at the apices to 2–3 µm wide, without or with a low notch. Asci 140–175 × 10–14 µm, 8-spored, unitunicate, cylindrical, hyaline, long pedicellate, arising from croziers, non-amyloid, ascospores released from an eccentric split at the apical apex. Ascospores narrowly fusoid, narrowed at both ends, inequilateral, with two large guttules, sometimes only with one big guttule, smooth, hyaline, (12.5–) 13–14.5 (–15) × (6–) 6.5–7 (–7.3) µm (L_m_ × W_m_ = 13.5 × 6.8 µm, Q = 1.9–2.1, Q_m_ = 2, n = 50). Receptacle surface with broad conical warts, 40–75 µm high, formed by short, fasciculate, hyphoid hairs, of 6–7 subglobose to elongated cells, constricted at septa, 6–10 µm wide. Resinous exudates abundant on the outer surface, dark yellow brown, partly dissolving and converting into small particles in MLZ, partially dissolving and turning yellowish brown in KOH. Basal mycelium of interwoven, 3–6 µm wide, septate, hyaline to pale brown hyphae, turning yellow in KOH, with abundant small, regularly arranged, spheroid, pale brown, resinous exudates, partly dissolving in KOH, unchanged in MLZ.

Other materials examined: China. Hebei Province, Chicheng County, Dahaituo nature reserve, alt. 1640m, on soil under *Betula platyphylla* Suk., 22 August 2019, J.Q. Li L482 (BJTC L482); China. Inner Mongolia Autonomous Region, Balinyou Banner, Saihanwula Nature Reserve, on soil in mixed forest of *Larix gmelinii* (Rupr.) Kuzen. and *Betula platyphylla* Suk., 2 September 2008, T.Z. Liu, H.M. Zhou & C. Sun 3858 (HMAS 188468).

Notes: *Otidea filiformis* diagnosed by the combination of yellowish-brown to ochre-yellow apothecia, sometimes with brown spots or stains, narrowly fusoid ascospores, uniform width, narrow paraphyses (≤3 µm) and basal mycelium with abundant spheroid, pale brown, resinous exudates. *O**tidea bufonia* and *O*. *mirabilis* are similar to *O. filiformis* in apothecia color and ascospore shape. *O**tidea bufonia* differs from *O. filiformis* in having the longer ascospores (12–) 13–16.5 (–18) × 6–7.5 (–8) µm, and brown striate exudates on some hyphae of the medullary excipulum. *O*. *mirabilis* differs in dark-brown apothecia, purple to lilaceous-bluish shades on the receptacle surface and biflabellate crystal-like exudates in the medullary excipulum. The apothecia of *Otidea korfii* Pfister, F. Xu & Z.W. Ge, *O. olivaceobrunnea* and *O. saliceticola* Cartabia, M. Carbone & P. Alvarado also have brown tones. *Otidea korfii* is distinguished from *O**. filiformis* by its ear-shaped apothecia, olivaceous brown receptacle surface, ellipsoid to broadly ellipsoid bigger ascospores (14.5–17 × 6.5–9 μm), resinous exudate of the ectal excipulum dissolving in MLZ and smooth basal mycelium. *Otidea olivaceobrunnea* differs from *O**. filiformis* in olive-brown hymenium, ellipsoid ascospores. *Otidea saliceticola* differs in pale alutaceous-greyish hymenium surface, dark brown receptacle surface, wider ascospores (14–15 × 7.5–8 μm) with low Q value of 1.75–1.87.

Phylogenetic analyses revealed that the sequences of *O. filiformis* were grouped into an independent clade with a strong support value (Figure 1 and Figure 2). DNA analysis showed that *O. filiformis* shared less than 95.88% ITS sequence similarity with other *Otidea* species. These supported the erection of the new species. One Finnish collection (MCVE 29372) identified as *O. bufonia* by Carbone et al. [22] is clustered into the *O. filiformis* clade with strong support values in our phylogenetic trees (Figure 1 and Figure 2), and it shared more than 98% ITS similarity with our *O. filiformis* and less than 94.6% with *O. bufonia*. This evidence showed MCVE 29372 is more closely related to *O. filiformis,* and whether it is conspecific with *O. filiformis* need further morphological identification.

***Otidea******khakicolorata*** L. Fan & Y.Y. Xu, sp. nov. (Figure 6).

MycoBank: MB843179.

Etymology: *khakicolorata*, referring to the khaki color of apothecia.

Holotype: China. Shanxi Province, Ningwu County, Guancen Mountain, Dashidong Forest Farm, alt. 2200m, among moss under coniferous forest dominated by *Picea wilsonii* Mast. and *Larix principis-rupprechtii* Mayr, 24 August 2017, X.Y. Yan YXY170824 (BJTC FM107).

Saprobic on soil. Apothecia solitary or gregarious in nature, 15–25 mm high, 5–8 mm wide, narrowly long ear-shaped, margin rounded, split, sessile or sub-stipitate. Hymenium surface khaki (#ca9a67) to pale ochre (#c08649) when fresh, yellowish ochre(#e3c57f) when dry, subsmooth. Receptacle surface concolorous with hymenium when fresh, slightly hygrophanous, dark reddish brown (#8c5738) when dry, furfuraceous. Warts absent or very low. Stipe, if present, very short. Basal tomentum and mycelium whitish to grayish whitish (#e7e2d9). Apothecial section 500–800 µm thick. Ectal excipulum of *textura angularis*, 70–120 µm thick, cells thin walled, brown, 10–25 × 6–20 µm. Medullary excipulum of *textura intricata*, 250–500 µm thick, hyphae 3.5–8 µm wide, sometimes slightly swollen, thin to slightly thick walled, septate, hyaline to light brown. Subhymenium ca. 60–90 µm thick, visible as a yellowish-brown zone, of densely arranged cylindrical to swollen cells. Paraphyses septate, curved to hooked, of uniform width at the apices, 2.5–3.7 µm wide, without notch. Asci 150–180 × 8.5–11 µm, 8-spored, unitunicate, cylindrical, hyaline, long pedicellate, arising from croziers, non-amyloid, ascospores released from an eccentric split at the apical apex. Ascospores ellipsoid, sometimes slightly inequilateral, with two large guttules, smooth, hyaline, (8.5–) 9–10 (–10.5) × (4.5–) 5–6 (–6.5) µm (L_m_ × W_m_= 9.5 × 5.5 µm, Q= 1.6–1.9, Q_m_= 1.75, n = 50). Receptacle surface with broadly conical warts, 30–40 µm high, formed by hyphoid hairs, of 2–5 subglobose to elongated cells, constricted at septa, 6–10 µm wide, sometimes with a gelatinous sheath. Resinous exudates abundant on the outer surface, yellow brown to dark brown, partly dissolving into amber drops in MLZ, turning reddish brown to dark reddish brown in KOH. Basal mycelium of 3–5.5 µm wide, septate, hyaline to pale-brown hyphae, unchanged in KOH, smooth or with little resinous exudates on the surface, partly dissolving in MLZ, and partly dissolving and more slowly in KOH.

Notes: *Otidea khakicolorata* is characterized by khaki to pale-ochre, long, narrowly ear-shaped apothecia, small ascospores and resinous exudates on the ectal excipulum turning reddish brown in KOH. *Otidea nannfeldtii* and *O.*
*khakicolorata* share similar apothecia shape and the reaction of the resinous exudate in the ectal excipulum and basal mycelium in MLZ and KOH, but *O. nannfeldtii* can be distinguished by an ochre to orangish-ochre hymenium surface with pink tones, higher warts (45–85 µm) on the apothecial outer surface, medullary excipulum of textura intricata differentiated into two parts, and relatively bigger ascospores ((9–) 9.5–10.5 (–11.5) × 5.5–6.5 (–7) µm). *Otidea*
*khakicolorata* is phylogenetically close to *O. nannfeldtii*; however, they are separated by a low support value (Figure 1 and Figure 2). DNA analyses showed that *O.*
*khakicolorata* shared less than 91% similarity in ITS sequence with *O. nannfeldtii*.

***Otidea parvula*** L. Fan & Y.Y. Xu, sp. nov. (Figure 7).

MycoBank: MB843180.

Etymology: *parvula*, referring to the small apothecia of the fungus.

Holotype: China. Shanxi Province, Jiaocheng County, Guandi Mountain, Pangquangou National Nature Reserve, alt. 2000m, on soil in the mixed forest dominated by *Picea wilsonii* Mast., 7 September 2017, J.Z. Cao, Cao170803 (BJTC FM210-A).

Saprobic on soil. Apothecia gregarious to caespitose in nature, 8–15 mm high, 5–13 mm wide, initially narrowly to broadly ear-shaped, margin rounded, then expanding and sometimes becoming irregularly ear-shaped or cup-shaped, split, stipitate, or sessile. Hymenium surface pale whitish ochre (#ffffed) to ochre yellow (#b39a7f) when fresh, pale ochre brown (#c2a461) when dry, subsmooth. Receptacle surface pale ochre (#d7c498) to orangish ochre (#dba35e) when fresh, hygrophanous, light yellowish brown (#d7c498) when dry, furfuraceous. Stipe 3–7 × 3–5 mm. Basal tomentum and mycelium white. Apothecial section 500–700 µm thick. Ectal excipulum of *textura angularis*, 70–120 µm thick, cells thin walled, pale brown, 9–24 × 7–20 µm. Medullary excipulum of *textura intricata*, 80–160 µm thick, hyphae 3–7 µm wide, sometimes slightly swollen, thin to thick walled, septate, hyaline to light brown. Subhymenium c. 50–80 µm thick, visible as a brown zone, of densely arranged cylindrical to swollen cells, with scattered brown resinous exudate at septa. Paraphyses septate, bent to curved, sometimes straight, of uniform width or slightly enlarged at the apices, 2.5–4.5 µm wide, sometimes with 1–2 notches near the apex. Asci 150–200 × 10.5–13.5 µm, 8-spored, unitunicate, cylindrical, hyaline, long pedicellate, arising from croziers, non-amyloid, ascospores released from an eccentric split at the apical apex. Ascospores ellipsoid, slightly inequilateral, with two large guttules, sometimes with only one big guttule, smooth, hyaline, (12.5–) 13–15.5 (–16) × (6.5–) 6.8–8 (–8.6) µm (L_m_ × W_m_ = 14.3 × 7.5 µm, Q = 1.7–2.1, Q_m_ = 1.9, n = 50). Receptacle surface with low warts, 30–50 µm high, formed by short, fasciculate, hyphoid hairs, of 5–6 subglobose to elongated cells, constricted at septa, 5–10 µm wide. Resinous exudates absent. Basal mycelium of 2.5–5 µm wide, septate, hyaline to pale brown hyphae, smooth, unchanged in KOH, turning yellow in MLZ.

Other materials examined: China. Shanxi Province, Jiaocheng County, Guandi Mountain, Pangquangou National Nature Reserve, alt. 2000m, on soil in the mixed forest dominated by *Picea wilsonii*, 7 September 2017, J.Z. Cao, Cao170803 (BJTC FM210-B).

Notes: *Otidea parvula* is easily recognized by the stipitate, irregularly ear-shaped or cup-shaped, small, pale-whitish-ochre, ochre-yellow to orangish-ochre apothecia, straight to bent paraphyses and furfuraceous receptacle surface. *Otidea parvula* and *O. adorniae* Agnello, M. Carbone & P. Alvarado are somewhat similar in the color of the apothecia, but *O. adorniae* differs in its larger apothecia and smaller ascospores (11.8 × 6.4 µm). *Otidea parvula* and *O. parvispora* (Parslow & Spooner) M. Carbone, Agnello, Kautmanová, Z.W. Ge & P. Alvarado have the highest ITS sequence similarity of 96%, but upon examination of the phylogenetic tree (Figure 1 and Figure 2), they don’t seem to be closely related, and *O. parvispora* is easily distinguished by its pale hymenium and smaller ascospores ((11.0–) 11.5–13.0 × 5.0–6.5 µm).

***Otidea plicara*** L. Fan & Y.Y. Xu, sp. nov. (Figure 8).

MycoBank: MB843181.

Etymology: *plicara*, referring to the small fold on the apothecia.

Holotype: China. Shanxi Province, Jiaocheng County, Guandi Mountain, Badaogou valley, alt. 2000m, on soil in the mixed forest dominated by *Picea wilsonii*, 7 September 2017, J.Z. Cao, Cao170855 (BJTC FM262-A).

Saprobic on soil. Apothecia solitary or gregarious in nature, 15–28 mm high, 12–42 mm wide, initially spoon-shaped or broadly ear-shaped, soon expending, in the end almost deeply cup-shaped, often broader above, margin entire, with a small fold on the apothecia, seemingly spilt yet not split, stipitate. Hymenium surface pale greyish brown (#6f6a61) to dark brown (#665856), margin yellow ochre (#837050) when fresh, when dry becoming slightly lighter but dull, light brown (#ab9876), subsmooth. Receptacle surface dark reddish brown (#7c6052) when fresh, slightly hygrophanous, pale ochre brown (#a48e6a) when dry, finely furfuraceous, wrinkle veined at the base. Stipe 8–15 × 5–8 mm. Basal tomentum and mycelium abundant, white to pale cream (#f1f9ed). Apothecial section 650–1000 µm thick. Ectal excipulum of *textura angularis*, 60–100 µm thick, cells thin walled, brown, 11–34 × 9–28 µm. Medullary excipulum of *textura intricata*, 300–500 µm thick, hyphae 3–7.5 µm wide, sometimes slightly swollen, thin walled, septate, hyaline to light brown. Subhymenium ca. 80–130 µm thick, visible as a yellowish-brown zone. Paraphyses septate, curved to hooked, usually enlarged at the apices, 3.5–5.5 µm wide at apex, 2–3 µm below. Asci 160–220 × 10–14 µm, 8-spored, unitunicate, cylindrical, hyaline, long pedicellate, arising from croziers, non-amyloid, ascospores released from an eccentric split at the apical apex. Ascospores ellipsoid, sometimes slightly inequilateral, with one to two large guttules, smooth, hyaline, (12.5–) 13.5–16 (–17) × (6–) 6.5–8 (–8.5) µm (L_m_ × W_m_ = 14.7 × 7.4 µm, Q = 1.8–2.2, Q_m_ = 2, n = 50). Receptacle surface with hyphoid hairs, 50–80 µm long, of 3–6 ovoid or subglobose to elongated cells, constricted at septa, 4–9 µm wide. Resinous exudates absent. Basal mycelium of interwoven, 2.5–6 µm wide, septate, hyaline to pale brown hyphae, unchanged in KOH, smooth, turning yellow in MLZ.

Other materials examined: China. Shanxi Province, Jiaocheng County, Guandi Mountain, Badaogou Scenic Area, alt. 2000m, on soil in the mixed forest dominated by *Picea wilsonii* Mast., 7 September 2017, J.Z. Cao, Cao170855 (BJTC FM262-B).

Notes: *Otidea plicara* is characterized by greyish-brown to dark-brown, stipitate, rarely split, deeply cup-shaped apothecia, small ascospores, enlarged paraphyses and the lack of resinous exudates on the ectal excipulum and basal mycelium. Macroscopically, *Otidea apophysata* (Cooke & W. Phillips) Sacc. and *O. platyspora* Nannf. have similar apothecial shape and color to *O. plicara*, but *O. apophysata* can be distinguished by the larger ascospores (20–24.5 × 9–11 µm) and frequently branched paraphyses. *Otidea platyspora* can be distinguished by split apothecia and larger ascospores (18–22 × (9.5–)10.5–12 µm). DNA analyses showed that *O*. *plicara* shared less than 92% similarity in ITS sequence with other species of *Otidea*. Phylogenetic analyses revealed that the sequences of *O. plicara* were grouped into an independent clade with a strong support value (Figure 1 and Figure 2). These supported the erection of the new species.

***Otidea******purpureobrunnea*** L. Fan & Y.Y. Xu, sp. nov. (Figure 9).

MycoBank: MB843182.

Etymology: *purpureobrunnea*, referring to the purple-brown tone of apothecia.

Holotype: China. Shanxi Province, Qinshui County, Tuwo Township, Shangwoquan Village, alt. 1200m, on soil under *Quercus* sp., 25 August 2020, H. Liu 1065 (BJTC FM1048).

Saprobic on soil. Apothecia gregarious to caespitose in nature, 25–55 mm high, 50–80 mm wide, initially ear shaped, soon expanding, becoming broadly ear shaped or deeply cup shaped, often elongated on one side, split, margin sometimes lobate, stipitate or sessile. Hymenium surface ochraceous brown (#804618), grayish purple (#5e4f5f) to purple brown (#39242f) when fresh, gray brown to dark brown (#3e2c1c) when dry, subsmooth. Receptacle surface grayish purple brown (#816e71) to dark purple brown (#483131) when fresh, sometimes partly dark yellow brown, slightly hygrophanous, some apothecia with shallowly wrinkled, dark brown (#492615) when dry, furfuraceous to finely warty. Stipe 5–10 × 4–8 mm. Basal tomentum and mycelium whitish to pale brown (#dccdbf). Apothecial section 900–1300 µm thick. Ectal excipulum of *textura angularis*, 80–120 µm thick, cells thin walled, brownish, 13–33 × 7–26 µm. Medullary excipulum of *textura intricata*, 500–900 µm thick, formed of loosely woven cylindrical to slightly swollen thin-walled hyphae, 4.5–11 µm wide, septate, hyaline to light brown, with brown resinous exudates at septa. Subhymenium c. 100–150 µm thick, visible as a brown zone, of densely arranged cylindrical to swollen cells, with scattered brown resinous exudate at septa. Paraphyses septate, curved to hooked, a few curved, sometimes forming a coil or helix, of the same width or often enlarged at the apices, 3.5–5 µm wide, 2–3.3 µm below, sometimes with 1–2 notches, or with an obvious bulge near the apex. Asci 140–190 × 8.5–15 µm, 8-spored, unitunicate, cylindrical, hyaline, long pedicellate, arising from croziers, non-amyloid, ascospores released from an eccentric split at the apical apex. Ascospores ellipsoid to slightly subfusoid, inequilateral, with two large guttules, sometimes with only one big guttule, smooth, hyaline, (12.5–) 13–15 (–15.5) × (6–) 6.5–7 (–7.5) µm (L_m_ × W_m_= 14 × 6.5 µm, Q= 1.9–2.3, Q_m_= 2.1, n = 50). Receptacle surface with broad conical warts, 35–60 µm high, formed by short, fasciculate, hyphoid hairs, of 2–5 subglobose to elongated cells, constricted at septa, 6–13 µm wide. Resinous exudates abundant on the outer surface, yellow brown to dark brown, partly dissolving into particles in MLZ, entirely dissolving and turning yellow in KOH. Basal mycelium of 3.5–6 µm wide, septate, hyaline to pale brown hyphae, turning yellow in KOH, mostly smooth, a few with very small, spheroid, pale-brown, resinous exudates, dissolving in KOH, partially dissolving in MLZ.

Other materials examined: China. Shanxi Province, Qinshui County, Tuwo Township, Shangwoquan Village, alt. 1200m, on soil under *Quercus* sp., 25 August 2020, H. Liu 1079 (BJTC FM1061).

Notes: *Otidea purpureobrunnea* is characterized by the stipitate, broadly ear-shaped to cup-shaped, grayish-purple to purple-brown apothecia, ellipsoid to slightly subfusoid ascospores, paraphyses enlarged at the apices, with 1–2 notches or an obvious bulge and smooth basal mycelium. Similar to *O. purpureobrunnea*, the apothecia of *O. bufonia, O. cupulata,*
*O. mirabilis*, *O. purpurea*, *O. purpureogrisea*, *O. smithii*, and *O. subpurpurea* all have some purple tones, but *Otidea bufonia* differs in its fusoid ascospores, the presence of hyphae with striate resinous exudates in the medullary excipulum, resinous exudates of the ectal excipulum not turning bright yellow in KOH, and in having abundant resinous exudates on the basal mycelium. *Otidea mirabilis* differs by having fusoid ascospores, resinous exudates of the ectal excipulum that do not turn bright yellow in KOH and when present, biflabellate, crystal-like exudates in the medullary excipulum. *Otidea purpurea* and *O. subpurpurea* are easily distinguished by the obviously smaller spores (*O. subpurpurea*: 9–12 × 4.5–6 µm; *O. purpurea*: 8–10 × 4.5–6 µm). *Otidea purpureogrisea* is distinguished by the purple-gray tone of the receptacle surface near the base and resinous exudates of the ectal excipulum turning amber in MLZ and turning brown in KOH. *Otidea smithii* is distinguished by typically narrower, ear-shaped apothecia, relatively shorter ascospores (12–14 × 6–7.5 µm) with a lower Q_m_ value (1.9–2), and resinous exudates of the ectal excipulum not turning bright yellow in KOH. For a comparison with *O. cupulata* see under that species below.

Phylogenetic analyses revealed that *O. purpureobrunnea* and *O. filiformis* are grouped together with a low support value (Figure 2), but *O. filiformis* is easy to distinguish from *O. purpureobrunnea* by its apothecia without purple tones, fusoid ascospores, same width and narrow paraphyses (≤3 µm), as well as its basal mycelium with abundant spheroid, pale brown, resinous exudates. DNA analysis showed that *O.*
*purpureobrunnea* shared less than 94.53% similarity in its ITS sequence with *O. filiformis*. These indicate that they are two different species.

***Otidea subpurpurea*** W.Y. Zhuang, Mycologia Montenegrina 10: 238 (2007).

Holotype: China, Yunnan Province, Kunming City, Kunming Institute of Botany, alt. 1980m, 8 October 2005, Z.L. Yang 4602, (HKAS 49443); Isotype (HMAS 97530).

= *Otidea bicolor* W.Y. Zhuang & Zhu L. Yang, Mycotaxon 112: 35 (2010).

Holotype: China, Yunnan Province, Kunming City, Heilongtan Park, 16 August 2008, Z.L. Yang 5156, (HKAS 54453); Isotype (HMAS 188415).

= *Otidea pruinosa* Ekanayaka, Q. Zhao & K.D. Hyde, Fungal Diversity 87: 130 (2017).

Holotype: China, Yunnan Province, Kunming City, Xishan Scenic Area, 15 September 2012, T. Guo 617, (HKAS 81819).

Materials examined: China, Yunnan Province, Kunming City, Kunming Institute of Botany, alt. 1980m, 9 October 2005, Z.L. Yang 4602, (HKAS 49443); Isotype (HMAS 97530). China, Yunnan Province, Kunming City, Heilongtan Park, 16 August 2008, Z.L. Yang 5156, (HKAS 54453); Isotype (HMAS 188415). ibid., (HKAS 54449). China, Yunnan Province, Kunming City, Xishan Scenic Area, 15 September 2012, T. Guo 617, (HKAS 81819).

Notes: *Otidea bicolor*, *O. pruinosa* and *O. subpurpurea* are highly similar species. In fact, previous scholars have also noticed the phenomenon that the type sequences of the three species are clustered together [22,25]; however, due to the unavailability of specimens, this issue has not been formally addressed. In this study, we examined the type specimens and obtained multiple loci sequences from them. DNA analyses revealed that *O. pruinosa*, *O. bicolor*, and *O. subpurpurea* share high sequence similarity (ITS: >98.87%; nrLSU: >99.53%; *tef1-α*: >99.72%; *rpb2*: >99.45%). We performed morphological observation on these type specimens and found that there was no obvious difference in microscopic features. The reaction of the resinous exudate in the ectal excipulum and basal mycelium in MLZ and KOH are also the same. Although the receptacle surface of *O. bicolor* and *O. subpurpurea* is purplish in tint when fresh, the receptacle surface of *O. pruinosa* is without a purplish tint [23,28,29], but that may be influenced by its habitat. *Otidea pruinosa* is proposed as a new species because of receptacle surface with pruinose, but we found a similar granulate on the surface of dry specimens of *O. bicolor* and *O. subpurpurea*. In addition, phylogenetic analyses based on the two-gene and four-gene datasets also confirmed that they represent the same species, so here we formally treat *O. bicolor* and *O. pruinosa* as synonyms of *O. subpurpurea*. The sequence from ZMU124 (label as *O. bufonia*) from Guizhou province of China grouped with *O. subpurpurea* with a high support value (Figure 1), indicating that *O. subpurpurea* seems widely distributed in southwest China. Similarly, the sequence from JS150904-08 from Korea named *O. bufonia* [45] was also grouped into this clade (Figure 1). We checked the original morphological description by Jin et al. [45] and found that its ascospores size does not conform to *O. bufonia*, but instead to *O. subpurpurea*. This indicates that *O. subpurpurea* also occurs in Korea.

***Otidea mirabilis*** Bolognini & Jamoni in Jamoni, Funghi e Ambiente 85–86: 56 (2001). (Figure 10).

Habitat: on soil under mixed forest of *Larix principis-rupprechtii* and *Betula* sp.

Distribution: Known from the northeast, northern, northwest and southwest regions of China.

Materials examined: China, Yunnan Province, Jingdong County, Ailao Mountain, Xujiaba Village, alt. 2500m, 24 August 1994, M. Zang, 12389 (HKAS 28129). China, Gansu Province, Wudu County, Liangshui Town, Gongba River Beach, alt. 2600m, 11 July 1996, M.S. Yuan, 2213 (HKAS 30708). China, Sichuan Province, Hongyuan County, Kangle Town, alt. 3400m, 19 August 1998, M.S. Yuan, 3433 (HKAS 33633). China, Jilin Province, Fusong County, Songjiang River, 19 August 2000, M.S. Yuan, 4725 (HKAS 37272). China, Xinjiang Autonomous Region, Jimusa’er, alt. 1700m, 1 August 2003, W.Y. Zhuang & Y. Nong, 4657 (HMAS 83568). China, Inner Mongolia Autonomous Region, Chifeng City, Baiyin Aobao National Nature Reserve, 2 August 2013, Tolgor Bau (HMJAU 26926). China, Shanxi Province, Jiaocheng County, Guandi Mountain, Shanshui Village, alt. 1800m, 8 September 2017, J.Z. Cao, CAO170863 (BJTC FM292). China, Shanxi Province, Jiaocheng County, Pangquangou Nature Reserve, alt. 2100m, 28 August 2018, H. Liu, LH234 (HSA 234).

Notes: The occurrence of *O. mirabilis* is confirmed in China based on morphological and DNA evidence in this study. Olariaga et al. [4] showed already that *O. mirabilis* occur in China using nrLSU sequences from two Chinese collections in GenBank (identified as *O. leporina* by Zhuang [17] and Liu and Zhuang [18], but by morphology it has not been previously confirmed, as Olariaga et al. did not study those two collections morphologically. It is interesting that two distinct clades were revealed, one comprising Chinese specimens, and another comprising specimens from Europe. The Chinese specimens shared 98.46–99.84% ITS sequence similarity and the European ones had 99.54–99.85% similarity, while the similarities between the two proveniences were 97–98.5%. However, we found no significant morphological differences between the Chinese and European specimens, which probably resulted from the geographic distance.

***Otidea nannfeldtii*** Harmaja, Karstenia 15: 31 (1976). (Figure 10).

Habitat: on soil under mixed forest of *Larix principis-rupprechtii*.

Distribution: Known in northern China and northwest China.

Materials examined: China, Shanxi Province, Ningwu County, Guancen Mountain, Qiuqiangou Village, alt. 2100m, on soil under *L. principis-rupprechtii* Mayr, 25 August 2017, X.Y. Yan, YXY170836 (BJTC FM168); ibid., X.Y. Yan, YXY170837 (BJTC FM169); ibid., X.Y. Yan, YXY170838 (BJTC FM170). China, Shanxi Province, Jiaocheng County, Guandi Mountain, Pangquangou Nature Reserve, alt. 2000m, 6 September 2017, J.Z. Cao, CAO170829 (BJTC FM236); ibid., J.Z. Cao, CAO170836 (BJTC FM243). China, Xinjiang Autonomous Region, Jimusa’er, alt. 1700m, 1 August 2003, W.Y. Zhuang & Y. Nong, 4655 (HMAS 83573).

Notes: The occurrence of *O. nannfeldtii* in China is first confirmed based on molecular and morphological evidence. *Otidea nannfeldtii* is originally described in Europe, and also reported from North America [4]. Before this study, there are no DNA data that support the existence of this species in China.

## 4. Discussion

Temperate China is surely rich in *Otidea* species. Nine species are added to this genus by this study. A total of 31 species is thus recorded in this huge country currently. Of these species, 27 species are supported by morphological and molecular data, but four species (*O. cochleata* (L.) Fuckel, *O. purpurea* (M. Zang) Korf & W.Y. Zhuang, *O. smithii*, *O. tianshuiensis* J.Z. Cao, L. Fan & B. Liu) still lack DNA evidence. Compared to the records from the continents of Europe (c. 32 accepted species) and North America (c. 14 accepted species), more studies of this large and widely distributed temperate fungal group in China are needed. From the present point of view, the *Otidea* species is widely distributed in the southwest and northern regions of China. Four species, *O. alutace**a*, *O. bufonia*, *O. mirablilis*, and *O. onotica*, are widely distributed and are often encountered in the wild. So far, almost no *Otidea* species have been reported from south-central China and east China, which also have abundant forest resources, so it is necessary to investigate fungal resources in these regions in the future.
Key to species of *Otidea* in the study1. Apothecia entire-----------------------------------------------------------------------------------------21. Apothecia split-------------------------------------------------------------------------------------------32. Apothecia broadly cup-shaped, ochre brown to reddish brown, with abundant resinous exudates on basal mycelium and ectal excipulum, ascospore lenth > 18 µm--------------------------------------------------------------------------------------------------------------------*O. propinquata*2. Apothecia deeply cup-shaped, with a small fold, pale greyish brown to dark brown, ectal excipulum and basal mycelium without resinous exudates, ascospore lenth ≤17 μm-----------------------------------------------------------------------------------------------------------*O. plicara*3. Apothecia long, narrowly ear-shaped-------------------------------------------------------------43. Apothecia cup-shaped or broadly ear-shaped---------------------------------------------------64. Apothecia khaki to pale ochre, receptacle surface with warts of 30–40 µm high, ascospores (8.5–) 9–10 (–10.5) × (4.5–) 5–6 (–6.5) µm ------------------------------------*O. khakicolorata*4. Apothecia cinnamon brown or yellowish ochre to brown, receptacle surface with warts of 45–85 µm high---------------------------------------------------------------------------------------------55. Ascospore length < 12 µm, resinous exudates of the ectal excipulum turning reddish brown in KOH------------------------------------------------------------------------------*O. nannfeldtii*5. Ascospore length > 12 µm, resinous exudates of the ectal excipulum turning yellowish reddish grey heterogeneous drops in KOH----------------------------------------------*O. leporina*6. Apothecia pale yellow, yellowish brown, ochraceous yellow, ochre orange------------76. Apothecia brown, dark brown, dark reddish brown or purple brown-------------------107. Apothecia small-sized (<1.5cm), ascospores (12.5–) 13–15.5 (–16) × (6.5–) 6.8–8 (–8.6) µm---------------------------------------------------------------------------------------------------------*O. parvula*7. Apothecia big-sized (>2cm)---------------------------------------------------------------------------88. Resinous exudates of the ectal excipulum and basal mycelium absent----------*O. asperior*8. Resinous exudates of the ectal excipulum and basal mycelium present------------------99. Hymenium light yellow to ochraceous yellow, often with pink tones, ascospore length < 11 µm-----------------------------------------------------------------------------------------*O. brevispora*9. Hymenium yellow to dull yellow, ascospore length > 11 µm----------------------*O. onotica*10. Resinous exudates of the ectal excipulum and basal mycelium absent-------*O. alutacea*10. Resinous exudates of the ectal excipulum and basal mycelium present----------------1111. Apothecia yellowish brown, without purple tones-------------------------------*O. filiformis*11. Apothecia yellowish brown to dark brown, or purple brown, with purple to lilaceous-bluish tones--------------------------------------------------------------------------------------------------1212. Ascospores narrowly fusoid-----------------------------------------------------------------------1312. Ascospores ellipsoid to slightly subfusoid-----------------------------------------------------1413. Receptacle surface mostly without purple tones, medullary excipulum with striate exudates covering some hyphae------------------------------------------------------------------*O. bufonia*13. Receptacle surface strikingly purple-violaceous (fresh); medullary excipulum without, or rarely with flabellate crystal-like exudates, forming cross-like aggregates---------------------------------------------------------------------------------------------------------------------*O. mirabilis*14. Ascospore length <12 µm-----------------------------------------------------------*O.subpurpurea*14. Ascospore length >12 µm---------------------------------------------------------------------------1515. Receptacle surface with purple tones, grayish purple brown to dark purple brown, basal mycelium smooth, or with a few pale brown resinous exudates---------------------------------------------------------------------------------------------------------------------*O. purpureobrunnea*15 Receptacle surface without purple tones, yellowish brown to brown, basal mycelium with abundant resinous exudates ----------------------------------------------------------*O. cupulata*

## Figures and Tables

**Figure 1 jof-08-00272-f001:**
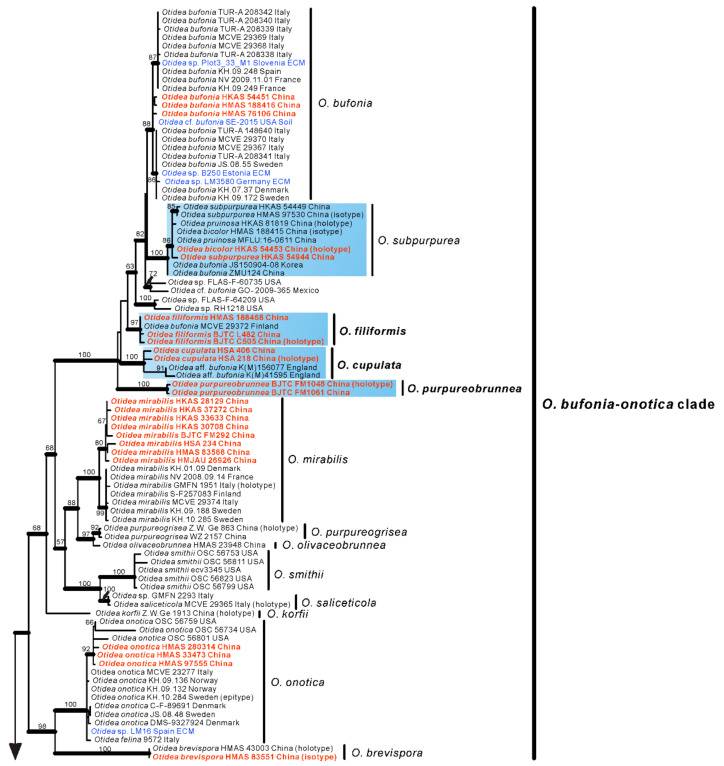
Phylogenetic tree generated from maximum likelihood analysis based on ITS and nrLSU sequences, showing the phylogenetic relationships of *Otidea*. *Monascella botryosa* and *Warcupia terrestris* are the outgroups. Maximum likelihood bootstrap support values (≥50%) are indicated above the nodes as BS. Thick black branches received Bayesian posterior probabilities (BPP) ≥ 0.95. Novel sequences are printed in bold red. Mycorrhizal or environmental sequences are printed in blue. The new species are in bold font and highlighted by blue boxes.

**Figure 2 jof-08-00272-f002:**
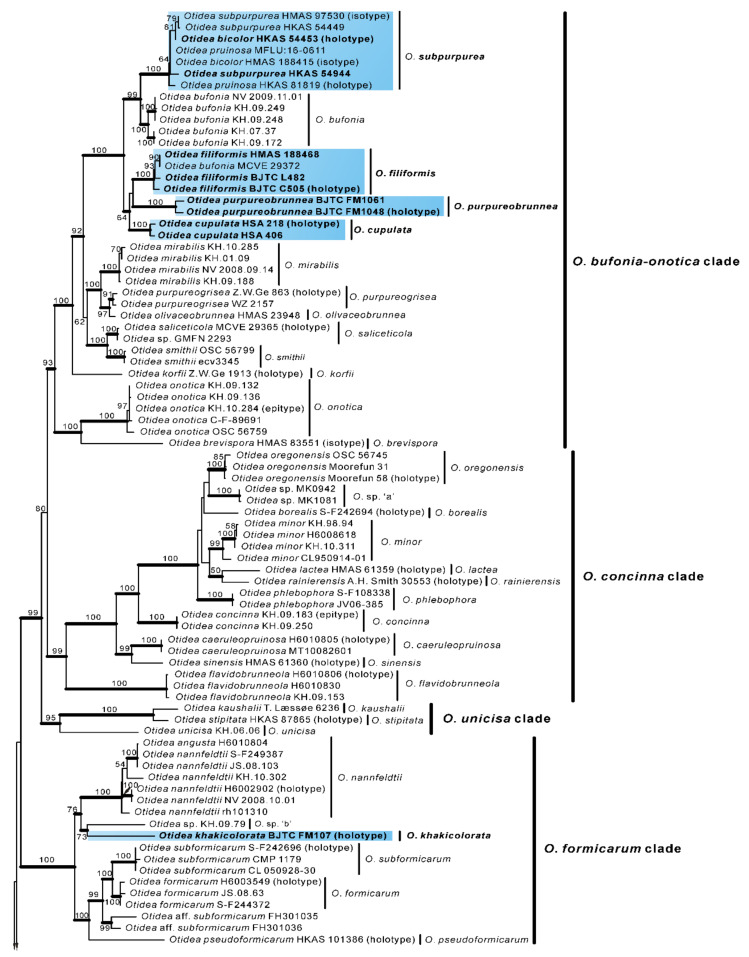
Phylogenetic tree generated from a maximum likelihood analysis based on ITS, nrLSU, *tef1-α* and *rpb2* sequences, showing the phylogenetic relationships of *Otidea*. *Monascella botryosa* and *Warcupia terrestris* are the outgroups. Maximum likelihood bootstrap support values (≥50%) are indicated above the nodes as BS. Thick black branches received Bayesian posterior probabilities (BPP) ≥ 0.95. The new species are in bold font and highlighted by blue boxes.

**Figure 3 jof-08-00272-f003:**
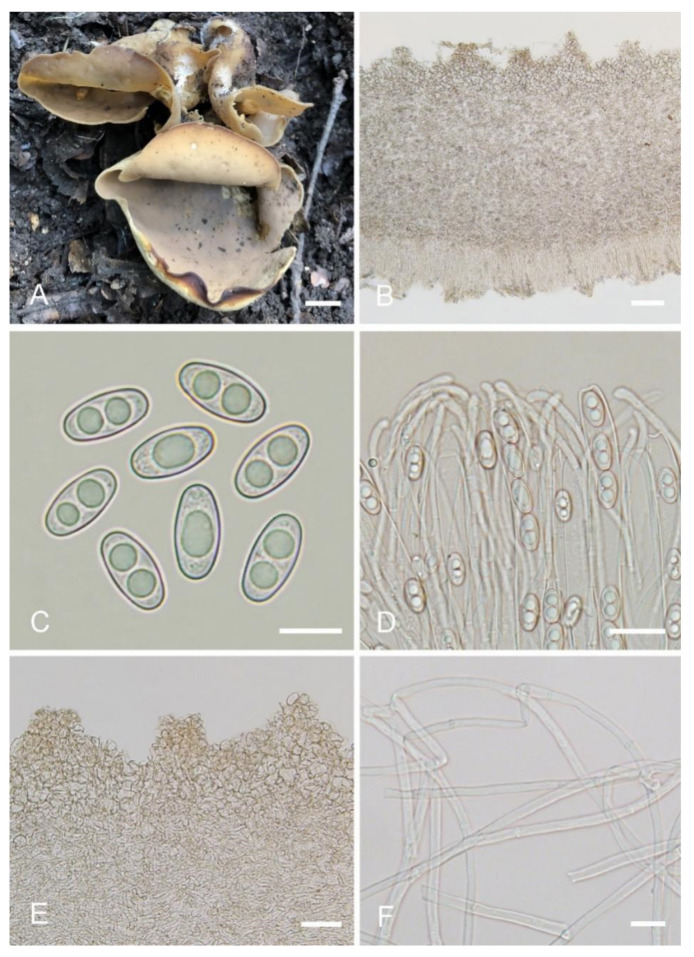
*Otidea aspera* (HSA 278) (**A**) apothecia, (**B**) anatomy of apothecium, (**C**) ascospores, (**D**) asci and paraphyses, (**E**) ectal excipulum in water, and (**F**) basal mycelium. Scale bars: (**A**) = 1 cm, (**B**) = 100 μm, (**C**) = 10 μm; (**D**) = 20 μm; (**E**) = 50 μm; (**F**) = 10 μm.

**Figure 4 jof-08-00272-f004:**
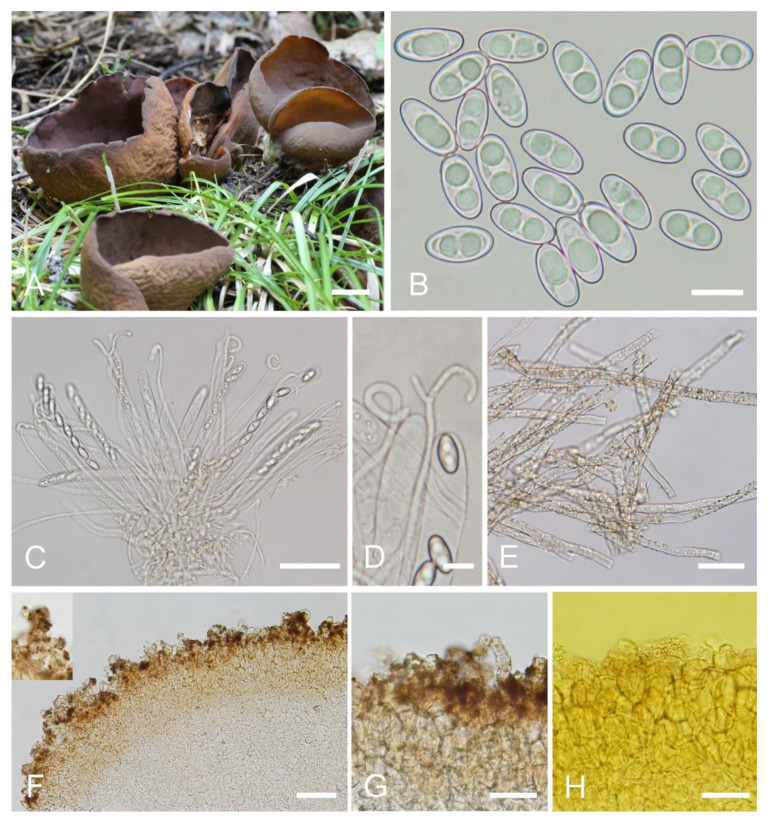
*Otidea cupulata* (HSA 218) (**A**) apothecia, (**B**) ascospores, (**C**) asci and paraphyses, (**D**) paraphyses, (**E**) basal mycelium, and (**F**) ectal and medullary excipulum in water, (**G**) ectal excipulum in water, (**H**) ectal excipulum in MLZ. Scale bars: (**A**) = 1 cm, (**B**) = 10 μm, (**C**) = 50 μm, (**D**) = 10 μm, (**E**) = 25 μm, (**F**) = 50 μm, (**G**,**H**) = 30 μm.

**Figure 5 jof-08-00272-f005:**
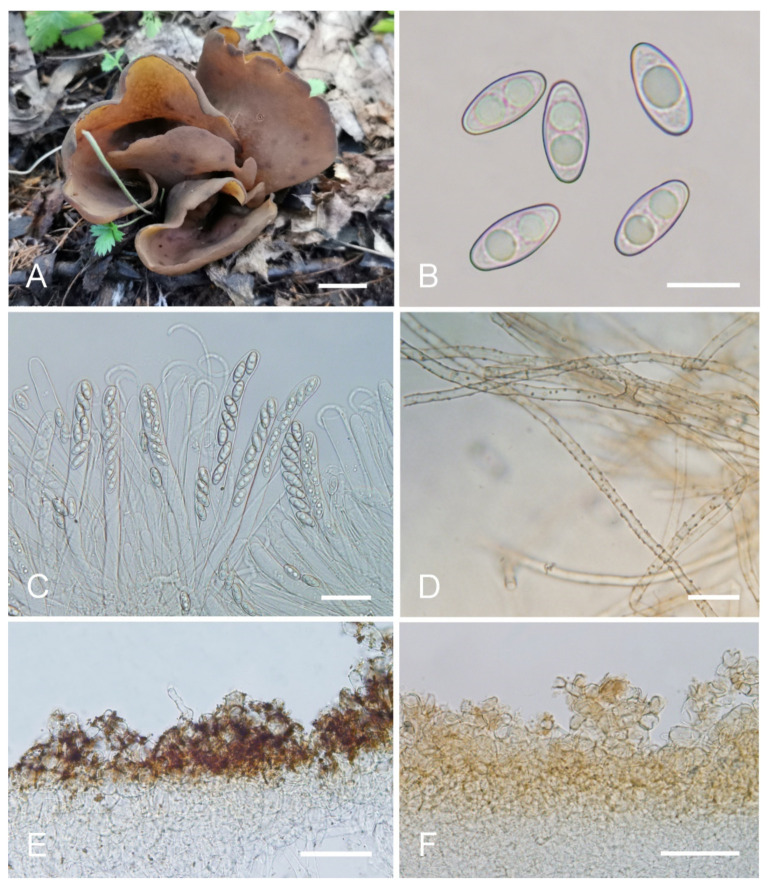
*Otidea filiformis* (BJTC C505) (**A**) apothecia, (**B**) ascospores, (**C**) asci and paraphyses, (**D**) basal mycelium, (**E**) ectal excipulum in water, and (**F**) ectal excipulum in KOH. Scale bars: (**A**) = 1 cm, (**B**) = 10 μm, (**C**) = 30 μm, (**D**) = 20 μm, (**E**,**F**) = 50 μm.

**Figure 6 jof-08-00272-f006:**
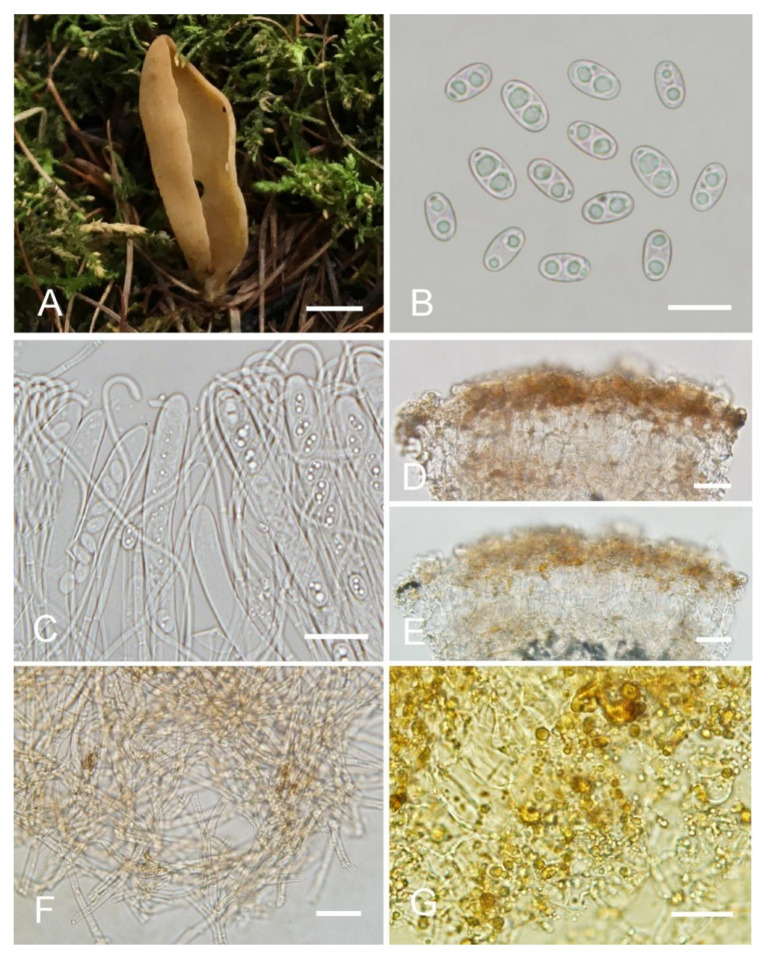
*Otidea khakicolorata* (BJTC FM107) (**A**) apothecia, (**B**) ascospores, (**C**) asci and paraphyses, (**D**) ectal excipulum in KOH, (**E**) ectal excipulum in water, (**F**) basal mycelium, and (**G**) amber drops on the outermost ectal excipulum cells in Melzer’s reagent. Scale bars: (**A**) = 0.5 cm, (**B**) = 10 μm, (**C**) = 20 μm, (**D**,**E**) = 25 μm, (**F**,**G**) = 20 μm.

**Figure 7 jof-08-00272-f007:**
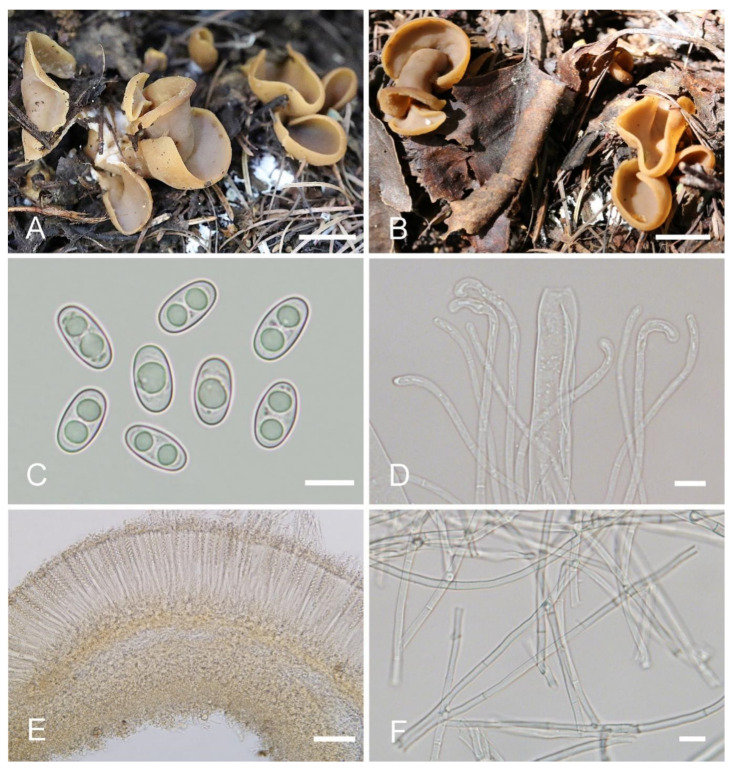
*Otidea parvula* (BJTC FM210-A) (**A**,**B**) apothecia, (**C**) ascospores, (**D**) paraphyses and asci, (**E**) anatomy of apothecium, and (**F**) basal mycelium. Scale bars: (**A**,**B**) = 1 cm, (**C**,**D**) = 10 μm, (**E**) = 100 μm, (**F**) = 10 μm.

**Figure 8 jof-08-00272-f008:**
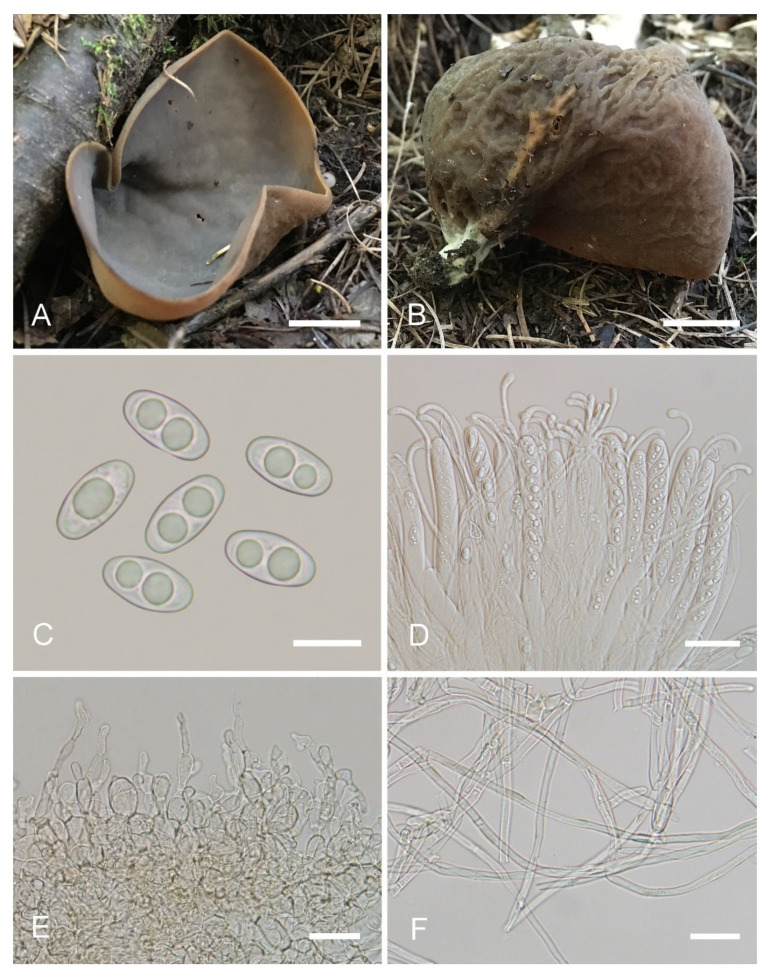
*Otidea plicara* (BJTC FM262-A) (**A**,**B**) apothecia, (**C**) ascospores, (**D**) paraphyses and asci, (**E**) ectal excipulum in water, and (**F**) basal mycelium. Scale bars: (**A**,**B**) = 1 cm, (**C**) = 10 μm, (**D**,**E**) = 30 μm, (**F**) = 20 μm.

**Figure 9 jof-08-00272-f009:**
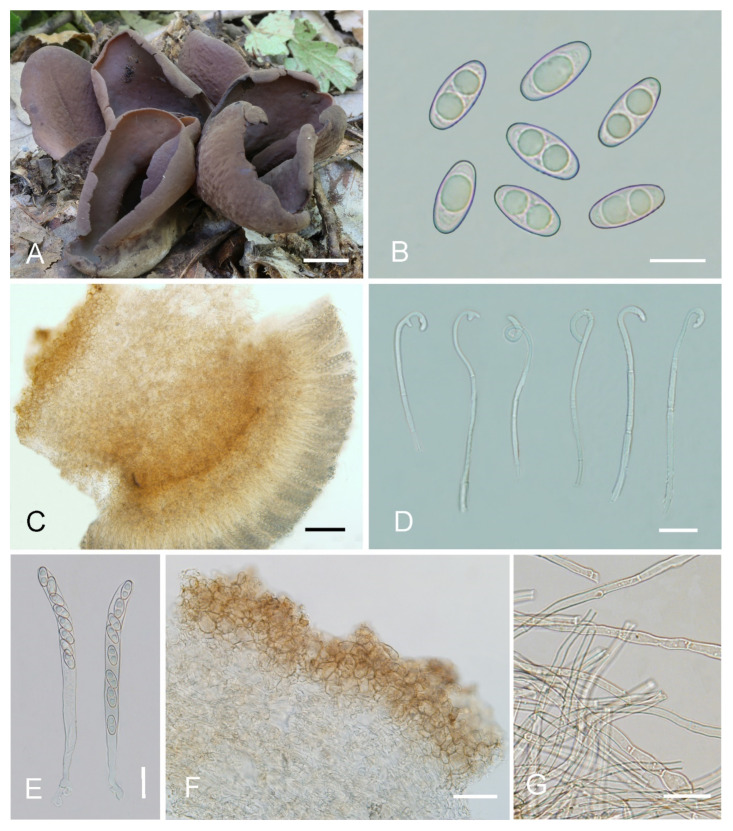
*Otidea purpureobrunnea* (BJTC FM1048) (**A**) apothecia, (**B**) ascospores, (**C**) anatomy of apothecium, (**D**) paraphyses, (**E**) asci, (**F**) ectal and medullary excipulum in water, and (**G**) basal mycelium. Scale bars: (**A**) = 1 cm, (**B**) = 10 μm, (**C**) = 100 μm, (**D**,**E**) = 20 μm, (**F**) = 50 μm, (**G**) = 20 μm.

**Figure 10 jof-08-00272-f010:**
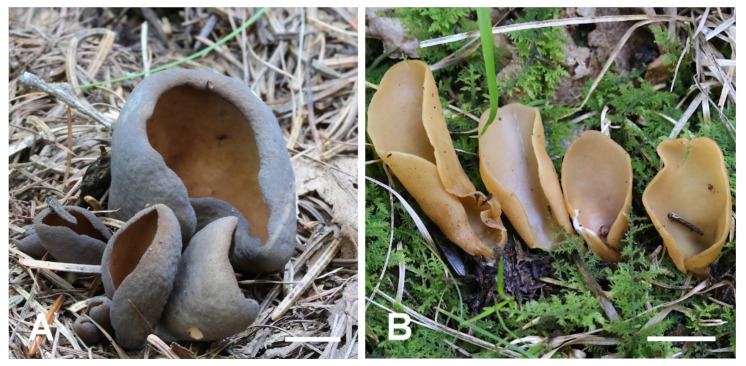
Two new record species from China. (**A**) *Otidea mirabilis* (HSA 234), (**B**) *Otidea nannfeldtii* (BJTC FM236). Scale bars: (**A**,**B**) = 1 cm.

## Data Availability

The sequencing data were submitted to GenBank.

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
