# Peer review of "New Species and New Records of Otidea from China Based on Molecular and Morphological Data"

_jof, 2022, doi:10.3390/jof8030272_

Round 1

Reviewer 1 Report

Review for Journal of Fungi of the manuscript: New species and new records of Otidea from China based on molecular and morphological data, by Yu-Yan Xu, Ning Mao, Jia-Jia Yang and Li Fan.

This is a valuable paper that is giving new information about Otidea in China. It is describing and illustrating 7 new species of Otidea and reporting an additional 2 species of Otidea as new to China, and confirming the occurrence of 3 other Otidea species in China using molecular data. Two species O. bicolor and O. pruinosa, are placed in synonymy with O. subpurpurea, based on sequences of the holotypes and morphology. All photo plates are important.

No specific species delimitation/recognition methods are employed, but the species are considered distinct when they in combined phylogenetic analyses of the datasets, (1) ITS-LSU rDNA, and (2) ITS-LSU-Tef1-rpb2, are forming separate well-supported groups or lineages. This is acceptable, especially since some species have only few known collections, but it could improve the paper to employ 1 or 2 molecular species delimitation methods. Also, even thought the Notes of each new species mention morphological features by which the species (easily) can be recognized, I am not fully convinced that for example all the species in the O. bufonia clade can be recognized by morphology.

Contrary to the monographic work of Otidea by Hansen and Olariaga (2015) and Olariaga et al. (2015), the authors have chosen to align ITS across all of Otidea and use ITS in both of the combined phylogenetic analyses of the whole genus. To be able to use ITS, parts of the ITS have been excluded. It should be specified to some degree what parts and how many nucleotides have been excluded (this information is missing). Looking at the alignment it is apparent that the many insertions between Otidea clades have been cut out in Se-Al by hand (as also said in the methods). However, some regions still appear ambiguously aligned (possibly not homologous) in ITS1 and ITS2. Alternatively, the authors could consider aligning the ITS-LSU rDNA for each of the well-known clades separately, including the new species, i.e. (1) the O. bufonia-onotica clade, (2) the O. cantharella clade, (3) the O. formicarum clade, or possible two clades together, like (4) O. alutaceae - O. platyspora clades that are monophyletic. Such ITS datasets of closely related species could be more reliably aligned and be analyzed with LSU to produce phylogenies of each clade. This is justifiable because the phylogeny of Otidea is well known and highly supported. In addition to these smaller trees of individual clade, a second dataset LSU-RPB2-EF1 would be supplementing, by giving information on the relationships among the clades within all of Otidea. This is a suggestion, since the EF1 and RPB2 regions have been sequenced from the new Chinese species.  

I assume that the different gene regions were analyzed separately, to test for conflict among the individual gene trees. Such information should be given in the Methods and Results.

The tree figures (Figs. 1 - 2) would be better presented using a larger font for the taxon names (so enlarging the trees too). It should be added in the legend that the new species are in bold font and highlighted by blue boxes.

Two of the new names proposed need to be evaluated. Otidea purpurabrunneae should not have the ending "ae" but "ea", and should be: Otidea purpureobrunnea, with "o" and not "a". Otidea splitopsis is not so nice, because it is a mixture of Latin and Greek. It might be that a name referring to another feature could be used.

In the Introduction, it is said that molecular techniques have revolutionized phylogenetics and species delimitation in Otidea. That is true, but a whole new set of morphological and histochemical features have also been introduced in Otidea by Harmaja (2009) and further by Hansen and Olariaga (2015) and Olariaga et al. (2015). These features have made it possible to recognize the species to a large extent morphologically and have since then been employed. It should be mentioned. These newly introduced features are also used in this current manuscript describing the species.

Authorities of taxon names are given in Introduction, L. 25, for Otidea and for Otidea onotica, but later authorities are not given. For example L. 160 gives O. leporina and O. propinquata for the first time, but the authors are missing. It is a taxonomic paper, so authors could be given the first time a taxon name is used. Otherwise the author of Otidea and O. onotica should be removed to be consistent. Also for example the authors of Pinus tabuliformis and Quercus wutaishansea are given (L. 217), so why not the Otidea species that are discussed.

Specific comments:

Introduction:

  1. 27. “Otidea species usually form ectomycorrhizae … “

Change to: Otidea species are considered to form ectomycorrhizae … “

It has not been shown that Otidea species usually form ectomycorrhizae and direct evidence is lacking for most species, but they are considered to do so (as discussed by Hansen and Olariaga 2015).

Taxonomy:

In many cases where the material is cited, the date and year is given before the habitat and host trees, but usually date and year would be given right before the collector, herbarium number and herbarium - so that the habitat information will follow the locality information.  

Generally in the text: replace "fruit bodies" / "fruiting body" with "apothecia", to be specific (no need to say fruit body). "Paraphyse" should be changed to "paraphyses" throughout the paper (as examples: L. 535. "... small spores, enlarged paraphyse, .. ",  and L. 539. "... and frequently branched paraphyse.").

  1. 295. Notes. O. cupularis is said to be easily recognized by the stipitate, cup-shaped, (split is missing here!), dark orange brown to dark purple brown hymenium etc. But O. bufornia was described by most of these features in Olariaga et al. (2015), like stipitate (stipe 5-14 x 7-10 mm), deeply cup-shaped, split, hymenium initially orange-brown, then dark orange brown, when dried greyish brown, slightly purple etc. and with spores size and Q values in the same range. In the Notes, O. cupularis is not compared morphologically to O. bufonia. The papers show that O. bufonia is present in China, so it would be useful to discuss how these species can be distinguished.

  1. 348. O. filiformis is discussed in relation to O. bufonia, but perhaps all of the species in the O. badia clade could be discussed together under one of these 3 new species (with a reference to those Notes under the 2 other species).

  1. 352-352. O. filiformis is said to be distinct (moreover) from O. bufonia by spheriod, pale brown, resinous exudates in the basal hyphae, but such resinous exudates also occur in O. bufonia.

  1. 398. Please go back to check the description of O. nannfeldtii in the monograph of Otidea, because O. khaki and O. nannfeldtii appear to share more features than the apothecial shape (if compared to the description given for O. khaki). Several Chinese collections of O. nannfeldtii were also studied in this manuscript and so, can be compared directly by the authors. I assume these were identical morphologically to the description in the monograph, since no comments are given on the morphology in the Taxonomy section (p. 22).

  1. 600. Olariaga et al. (2015, Fig. 1) showed already that O. mirabilis occur in China using sequences from GenBank (identified as O. leporina by Zhuang and Lio & Zhuang) - and Olariaga et al. also mentioned this in the Notes to O. mirabilis. But by morphology it has not been confirmed before, because Olariaga et al. did not study those two collections morphologically.

Reviewer 2 Report

Dear Authors

It was a pleasure to read your paper. Your descriptions and pictures are excellent. Nevertheless, the English language needs to be improved.

Some recommendations:

You do not give any information on the base of the asci: are there clamps/hooks or not – in other groups this feature is characteristic

It would be very helpful to add a key for the species you studied. This would significantly increase the relevance of your paper.

Author Response

The response to the reviewer’ comments

Dear reviewer,

Thank you for your comments and constructive suggestions concerning our manuscript (jof-1600181). Those comments are all valuable and very helpful for revising and improving our paper. We have studied comments carefully and have made corrections that we hope meet with approval. All the corrections and changes are marked in blue in the revised manuscript.

The following is a point-to-point response to the two reviewers’ comments.

It was a pleasure to read your paper. Your descriptions and pictures are excellent. Nevertheless, the English language needs to be improved.

Answer: Many thanks to the reviewer for your recognition. The English language has been polished, including grammar, words, and sentences, etc.

You do not give any information on the base of the asci: are there clamps/hooks or not – in other groups this feature is characteristic.

Answer: Thanks for the reviewer’s suggestions. The reviewer’s suggestions have been accepted. We have supplemented the characteristics of the asci of all species in manuscript. See line 253-255, 297-299, 302, 372-374, 444-446, 489-491, 537-539, 584-586.

It would be very helpful to add a key for the species you studied. This would significantly increase the relevance of your paper.

Answer: Thanks for the reviewer’s suggestions. The reviewer’s suggestions have been accepted. See line 728-775.

Reviewer 3 Report

Dear Authors,

The paper is unique and the detial data presented in the MS seem to warrant the novelty of the reported species.

However, there are some comments and suggestions on the MS in the attached file.

Thanks

Author Response

The response to the reviewer’ comments

Dear reviewer,

Thank you for your comments and constructive suggestions concerning our manuscript (jof-1600181). Those comments are all valuable and very helpful for revising and improving our paper. We have studied comments carefully and have made corrections that we hope meet with approval. All the corrections and changes are marked in blue in the revised manuscript.

The following is a point-to-point response to the two reviewers’ comments.

  1. Line 17-18. Did you check whether your new species are correct to Latin name?

Answer: Thanks for the reviewer’s suggestions. The reviewer’s suggestions have been accepted. We re-examined the Latin names of the new species and revised five of them. See line 17-18.

  1. Line Arrange in alphabetical order.

Answer: Thanks. Reviewer’s suggestions have been adopted. See line 22.

  1. Line Please add morphological characteristic of Otidea.

Answer: Thanks for the reviewer’s suggestions. The reviewer’s suggestions have been accepted. See line 26-30.

  1. Line 121-123.

Answer: Thanks for reminding. Thousand separator has been added. See line 133-134.

  1. Line Reference citation are missing.

Answer: Thanks for reminding. Literature citations have been added. See line 153.

  1. Figure 1. Add legend here and other add Cont. Please change light blue-geen color text to another color as it's difficult for reader.

Answer: Thanks for the reviewer’s suggestions. The reviewer’s suggestions have been accepted. We changed the colors to be easier to read. See Figure 1.

  1. Figure 2. Add legend here and other add Cont. Add full name of asperior. And check some clads you all full name like Otidea platyspora, Otidea subterr...

Answer: Thanks for the reviewer’s suggestions. The reviewer’s suggestions have been accepted. We have checked the whole figure 2 and made corrections. See Figure 2.

  1. Line Add number and I didn't see you mention how register MB number in M&M.

Answer: Thanks. We have added MB numbers for seven new species. See line 229, 277, 351, 421, 469, 514, 561.

  1. Line 233. Every species should start with "Saprobic on ......". And this difficult to follower description specially sectioning part so please rearrange from Apothecia, Hymenium, Receptacle, Disc , Ectal excipulum, Medullary excipulum, Paraphyses, Asci, Ascospores

Answer: Thanks for the reviewer’s suggestions. The reviewer’s suggestions have been accepted. We have rearranged the description order of the section part. See morphological description of each species.

  1. Line 234. Need more details; uni or bi tunicate, color, perdicel, amyloid or non-amyloid, ascospores release from which part of asci. Need to check all other species in manuscript

Answer: Thanks for the reviewer’s suggestions. The reviewer’s suggestions have been accepted. We have supplemented the characteristics of the asci of all species in manuscript. See line 253-255, 297-299, 302, 372-374, 444-446, 489-491, 537-539, 584-586.

  1. Line 237, 239. textura intricata, textura angularis. Italic, check though out manuscript.

Answer: Thanks for the reviewer’s suggestions, the reviewer’s suggestions have been accepted. We have checked the full text.

  1. Line 304. Better to compare morphology as well.

Line 307-308. If both morphology and phylogeny support you can mention this sentence.

Answer: Thanks for the reviewer’s suggestions. However, as these specimens were not described in detail in the articles, we cannot compare them morphologically with our species.So we modified the sentence. See line 345-349.

  1. Line 354. In Figs 1,2 can see that bufonia strain MCVE 29372 from Finland clustered together with your O. filiformis so this one you need to discuss why they group together? What wrong with O. bufonia strain MCVE 29372?

Answer: Thanks for the reviewer’s suggestions. We have supplemented the discussion about the specimen from Finland. Whether it is our O. filiformis is not morphologically certain since we do not exam the specimen. See line 410-415.

  1. Line 358. Ascospores, change though out manuscript

Answer: Thanks for the reviewer’s suggestions. The reviewer’s suggestions have been accepted. We have replaced spores with ascospores throughout the text.

  1. Line 625. Any suggestion on Otidea study in future ?

Answer: Thanks for the reviewer’s suggestions. The reviewer’s suggestions have been accepted. See line 725-727.

  1. Line 636. Make sure all sequences data submit and get Genbank number before online.

Answer: Thanks. We have added Genbank number to the supplementary material. See Table S1.

  1. Regarding errors in citation format and grammar writing, we have checked and revised the full text. Thanks to the reviewer for the correction.

Round 2

Reviewer 3 Report

Dear Authors,

Thanks for revising and considering my suggestions.